# Certified Training:
# Small Boxes are All You Need

**Mark Niklas Müller,** * **Franziska Eckert,** * **Marc Fischer & Martin Vechev**
Department of Computer Science
ETH Zurich, Switzerland
{mark.mueller,marc.fischer,martin.vechev}@inf.ethz.ch, eckertf@student.ethz.ch

## Abstract

We propose the novel certified training method, SABR, which outperforms existing methods across perturbation magnitudes on MNIST, CIFAR-10, and TINY-IMAGENET, in terms of both *standard and certifiable accuracies*. The key insight behind SABR is that propagating interval bounds for a small but carefully selected subset of the adversarial input region is sufficient to approximate the worst-case loss over the whole region while significantly reducing approximation errors. SABR does not only establish a new state-of-the-art in all commonly used benchmarks but, more importantly, points to a new class of certified training methods promising to overcome the robustness-accuracy trade-off.

## 1 Introduction

As neural networks are increasingly deployed in safety-critical domains, formal robustness guarantees against adversarial examples (Biggio et al., 2013; Szegedy et al., 2014) are more important than ever. However, despite significant progress, specialized training methods that improve certifiability at the cost of severely reduced accuracies are still required to obtain deterministic guarantees.

Generally, both training and certification methods compute a network's reachable set given an input region defined by an adversary specification and a concrete input, by propagating a symbolic over-approximation of this region through the network (Singh et al., 2018, 2019; Gowal et al., 2018a). Depending on the method used for propagation, both the computational complexity and tightness of this approximation can vary widely. For certified training, an over-approximation of the worst-case loss is computed from this reachable set and then optimized (Mirman et al., 2018; Zhang et al., 2020; Wong et al., 2018). Surprisingly, the least precise propagation methods yield the highest certified accuracies as more precise methods induce significantly harder optimization problems (Jovanovic et al., 2021). However, the large approximation errors incurred by these imprecise methods lead to over-regularization and thus poor accuracy. Combining precise worst-case loss approximations and a tractable optimization problem is thus the core challenge of certified training.

In this work, we tackle this challenge and propose a novel certified training method, SABR, **S**mall **A**dversarial **B**ounding **R**egions, based on the following key insight: by propagating small but carefully selected subsets of the adversarial input region with imprecise methods (i.e., BOX), we can obtain *both* well behaved optimization problems and precise approximations of the worst case loss. This yields networks with complex neuron interactions, enabling higher standard and certified accuracies, while pointing to a new class of certified training methods with significantly reduced regularization. SABR, thus, achieves state-of-the-art standard *and* certified accuracies across all commonly used settings on the MNIST, CIFAR-10, and TINYIMAGENET datasets.

---

*Equal contribution

2022 Trustworthy and Socially Responsible Machine Learning (TSRML 2022) co-located with NeurIPS 2022.

**Main Contributions**    Our main contributions are:

- A novel certified training method, SABR, reducing over-regularization to improve both standard and certified accuracy (§3).
- A theoretical investigation motivating SABR by deriving new insights into the growth of BOX relaxations during propagation (§4).
- An extensive empirical evaluation demonstrating that SABR outperforms *all* state-of-the-art certified training methods in terms of both *standard and certifiable accuracies* on MNIST, CIFAR-10, and TINYIMAGENET (§5).

## 2    Background

In this section, we provide the necessary background for SABR.

**Adversarial Robustness**    Consider a classification model $\boldsymbol{h}\colon \mathbb{R}^{d_{\text{in}}} \mapsto \mathbb{R}^c$ that, given an input $\boldsymbol{x} \in \mathcal{X} \subseteq \mathbb{R}^{d_{\text{in}}}$, predicts numerical scores $\boldsymbol{y} := \boldsymbol{h}(\boldsymbol{x})$ for every class. We say that $\boldsymbol{h}$ is adversarially robust on an $\ell_p$-norm ball $\mathcal{B}_p^{\epsilon_p}(\boldsymbol{x})$ of radius $\epsilon_p$ if it consistently predicts the target class $t$ for all perturbed inputs $\boldsymbol{x}' \in \mathcal{B}_p^{\epsilon_p}(\boldsymbol{x})$. More formally, we define *adversarial robustness* as:

$$\arg\max_j h(\boldsymbol{x}')_j = t, \quad \forall \boldsymbol{x}' \in \mathcal{B}_p^{\epsilon_p}(\boldsymbol{x}) := \{\boldsymbol{x}' \in \mathcal{X} \mid \|\boldsymbol{x} - \boldsymbol{x}'\|_p \leq \epsilon_p\}. \tag{1}$$

**Neural Network Verification**    To verify that a neural network $\boldsymbol{h}$ is adversarially robust, several verification techniques have been proposed.

A simple but effective such method is verification with the BOX relaxation (Mirman et al., 2018), also called interval bound propagation (IBP) (Gowal et al., 2018b). Conceptually, we propagate the input region $\mathcal{B}_p^{\epsilon_p}(\boldsymbol{x})$ in form of a hyper-box relaxation (each dimension is described as an interval) through the network to compute an over-approximation of its reachable set and then check whether all included outputs yield the correct classification. Given an input region $\mathcal{B}_p^{\epsilon_p}(\boldsymbol{x})$, we over-approximate it as a hyper-box, centered at $\bar{\boldsymbol{x}}^0 := \boldsymbol{x}$ and with radius $\boldsymbol{\delta}^0 := \epsilon_p$, such that we have the $i^{\text{th}}$ dimension of the input $\boldsymbol{x}_i^0 \in [\bar{x}_i^0 - \delta_i^0, \bar{x}_i^0 + \delta_i^0]$. Given a linear layer $\boldsymbol{f}_i(\boldsymbol{x}^{i-1}) = \boldsymbol{W}\boldsymbol{x}^{i-1} + \boldsymbol{b} =: \boldsymbol{x}^i$, we obtain the hyper-box relaxation of its output defined by center $\bar{\boldsymbol{x}}^i = \boldsymbol{W}\bar{\boldsymbol{x}}^{i-1} + \boldsymbol{b}$ and radius $\boldsymbol{\delta}^i = |\boldsymbol{W}|\boldsymbol{\delta}^{i-1}$, where $|\cdot|$ denotes the elementwise absolute value. A ReLU activation $\text{ReLU}(\boldsymbol{x}^{i-1}) := \max(0, \boldsymbol{x}^{i-1})$ can be relaxed by propagating the lower and upper bound separately, resulting in the output hyper-box with $\bar{\boldsymbol{x}}^i = \frac{\boldsymbol{u}^i + \boldsymbol{l}^i}{2}$ and $\boldsymbol{\delta}^i = \frac{\boldsymbol{u}^i - \boldsymbol{l}^i}{2}$ where $\boldsymbol{l}^i = \text{ReLU}(\bar{\boldsymbol{x}}^{i-1} - \boldsymbol{\delta}^{i-1})$ and $\boldsymbol{u}^i = \text{ReLU}(\bar{\boldsymbol{x}}^{i-1} + \boldsymbol{\delta}^{i-1})$. We can now show provable robustness if we find the upper bound on the logit difference $y_i^\Delta := y_i - y_t < 0, \ \forall i \neq t$ to be smaller than 0.

Beyond BOX, more precise verification approaches track more relational information at the cost of increased computational complexity (Palma et al., 2022; Wang et al., 2021; Ferrari et al., 2022).

**Training for Robustness**    For neural networks to be certifiably robust, special training is necessary. Given a data distribution $(\boldsymbol{x}, t) \sim \mathcal{D}$, standard training generally aims to find a network parametrization $\boldsymbol{\theta}$ that minimizes the expected cross-entropy loss:

$$\theta_{\text{std}} = \arg\min_\theta \mathbb{E}_\mathcal{D}[\mathcal{L}_{\text{CE}}(\boldsymbol{h}_\theta(\boldsymbol{x}), t)], \quad \text{with} \quad \mathcal{L}_{\text{CE}}(\boldsymbol{y}, t) = \ln\big(1 + \sum_{i \neq t} \exp(y_i - y_t)\big). \tag{2}$$

When training for robustness, we, instead, wish to minimize the expected *worst case loss* around the data distribution, leading to the min-max optimization problem:

$$\theta_{\text{rob}} = \arg\min_\theta \mathbb{E}_\mathcal{D}\big[ \max_{\boldsymbol{x}' \in \mathcal{B}_p^{\epsilon_p}(\boldsymbol{x})} \mathcal{L}_{\text{CE}}(\boldsymbol{h}_\theta(\boldsymbol{x}'), t)\big]. \tag{3}$$

Unfortunately, solving the inner maximization problem is generally intractable. Therefore, it is commonly under- or over-approximated, yielding adversarial and certified training, respectively.

**Adversarial Training**    Adversarial training optimizes a lower bound on the inner optimization objective in Eq. (3) by first computing concrete examples $\boldsymbol{x}' \in \mathcal{B}_p^{\epsilon_p}(\boldsymbol{x})$ that maximize the loss term and then optimizing the network parameters $\boldsymbol{\theta}$ for these samples. While networks trained this way typically exhibit good empirical robustness, they remain hard to formally verify and sometimes also vulnerable to stronger or different attacks (Tramèr et al., 2020; Croce & Hein, 2020).

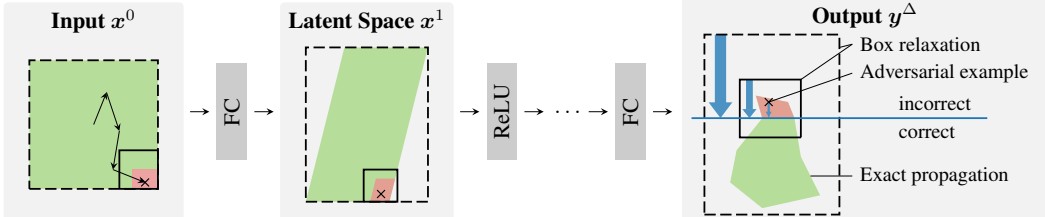

Figure 1: Illustration of SABR training. Instead of propagating a BOX approximation (dashed box ⬚) of the whole input region (red ▪ and green ▪ shapes in input space), SABR propagates a small subset of this region (solid box ▫), selected to contain the adversarial example (black ×) and thus the misclassified region (red). The smaller BOX accumulates much fewer approximation errors during propagation, leading to a significantly smaller output relaxation, which induces much less regularization (medium blue ⬇) than training with the full region (large blue ⬇), but more than training with just the adversarial example (small blue ⬇).

**Certified Training**  Certified training optimizes an upper bound on the inner maximization objective in Eq. (3), obtained via a bound propagation method. These methods compute an upper bound $u_{y^\Delta}$ on the logit differences $y^\Delta := y - y_t \mathbf{1}$ to obtain the robust cross-entropy loss $\mathcal{L}_{\text{CE,rob}}(\mathcal{B}_p^{\epsilon_p}(x), t) = \mathcal{L}_{\text{CE}}(u_{y^\Delta}, t)$. Surprisingly, using the imprecise BOX relaxation (Mirman et al., 2018; Gowal et al., 2018b) (denoted IBP) consistently produces better results than methods based on tighter abstractions (Zhang et al., 2020; Balunovic & Vechev, 2020; Wong et al., 2018). Jovanovic et al. (2021) trace this back to the optimization problems induced by the more precise methods becoming intractable to solve. While the heavily regularized, certifiably trained networks are amenable to certification, they suffer from severely reduced (standard) accuracies. Overcoming this robustness-accuracy trade-off remains a key challenge of robust machine learning.

## 3   Method – Small Regions for Certified Training

We address this challenge by proposing a novel certified training method, SABR — **S**mall **A**dversarial **B**ounding **R**egions — yielding networks that are amenable to certification and retain relatively high standard accuracies. We leverage the key insight that computing an over-approximation of the worst-case loss for a small but carefully selected subset of the input region $\mathcal{B}_p^{\epsilon_p}(x)$ often still captures the actual worst-case loss, while significantly reducing approximation errors.

We illustrate this in Fig. 1. Existing certified training methods propagate the whole input region (dashed box ⬚ in the input panel), yielding quickly growing approximation errors. The resulting imprecise over-approximations of the worst case loss (compare the red and green regions to the dashed box ⬚ in the output panel) cause significant over-regularization (large blue arrow ⬇). Adversarial training methods, in contrast, only consider individual points (× in Fig. 1) and fail to capture the worst-case loss, leading to insufficient regularization (small blue arrow ⬇ in the output panel). We tackle this problem by propagating small, adversarially chosen subsets of the input region (solid box ▫ in the input panel), which we call *propagation regions*. This yields significantly reduced approximation errors and thus more precise, although not necessarily sound over-approximation of the loss (see the solid box ▫ in the output panel). The resulting intermediate level of regularization (medium blue arrow ⬇) allows us to train networks that are both robust and accurate.

We observe that, depending on the size of the propagated region, SABR can be seen as a continuous interpolation between adversarial training for infinitesimally small regions and standard certified training for the full input region.

**Selecting the Propagation Region**  We parametrize the propagation region as an $\ell_p$-norm ball $\mathcal{B}_p^{\tau_p}(x')$ with center $x'$ and radius $\tau_p \leq \epsilon_p - \|x - x'\|_p$, ensuring that we indeed propagate a subset of the original region $\mathcal{B}_p^{\epsilon_p}(x)$. For notational clarity, we drop the subscript $p$. We first choose $\tau = \lambda\epsilon$ by scaling the original perturbation radius $\epsilon$ with the subselection ratio $\lambda \in (0, 1]$. We then select $x'$ by first conducting a PGD attack, yielding the preliminary center $x^*$, and then ensuring that the obtained region is fully contained in the original one by projecting $x^*$ onto $\mathcal{B}^{\epsilon-\tau}(x)$ to obtain $x'$. We show this in Fig. 2.

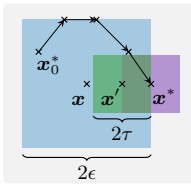

Figure 2: Illustration of SABR's propagation region selection process.

**Propagation Method**   While SABR can be instantiated with any certified training method, we chose BOX propagation (DIFFAI Mirman et al. (2018) or IBP (Gowal et al., 2018b)) to obtain well-behaved optimization problems (Jovanovic et al., 2021).

## 4   Understanding SABR – Robust Loss and Growth of Small Boxes

In this section, we aim to uncover the reasons behind SABR's success. Towards this, we first analyze the relationship between robust loss and over-approximation size before investigating the growth of the BOX approximation with propagation region size.

**Robust Loss Analysis**   Certified training typically optimizes an over-approximation of the worst-case cross-entropy loss $\mathcal{L}_{\text{CE,rob}}$, computed via the softmax of the upper-bound on the logit differences $\boldsymbol{y}^\Delta := \boldsymbol{y} - y_t$. When training with the BOX relaxation and assuming the target class $t$, w.l.o.g., we obtain $\boldsymbol{y}^\Delta \in [\bar{\boldsymbol{y}}^\Delta - \boldsymbol{\delta}^\Delta, \bar{\boldsymbol{y}}^\Delta + \boldsymbol{\delta}^\Delta]$ and the robust cross entropy loss $\mathcal{L}_{\text{CE, rob}}(\boldsymbol{x}) = \ln\left(1 + \sum_{i=2}^{n} e^{\bar{y}_i^\Delta + \delta_i^\Delta}\right)$. Further, we note that the BOX relaxations of many functions preserve the box centers, i.e., $\bar{\boldsymbol{x}}^i = \boldsymbol{f}(\bar{\boldsymbol{x}}^{i-1})$. Only unstable ReLUs, i.e., ReLUs containing 0 in their input bounds, introduce a slight shift. However, these are empirically few in certifiably trained networks (see Table 5). We can thus decompose the logit differences determining the robust loss into an accuracy term $\bar{\boldsymbol{y}}^\Delta$, corresponding to the misclassification margin of the adversarial example $\boldsymbol{x}'$ at the center of the propagation region, and a robustness term $\boldsymbol{\delta}^\Delta$, bounding the difference to the actual worst-case logits. As these terms generally represent conflicting objectives, robustness and accuracy are balanced to minimize the robust optimization objective. Consequently, reducing the regularization induced by the robustness term will bias the optimization process towards (standard) accuracy.

**BOX Growth**   We investigate the growth of BOX relaxations for an $L$-layer network with linear layers $\boldsymbol{f}_i$ and ReLU activation functions $\boldsymbol{\sigma}$. Given a BOX input with radius $\delta^{i-1}$ and center distribution $\bar{x}^{i-1} \sim \mathcal{D}$, we define the per-layer growth rate $\kappa^i = \frac{\mathbb{E}_\mathcal{D}[\delta^i]}{\delta^{i-1}}$ as the ratio of input and expected output radius.

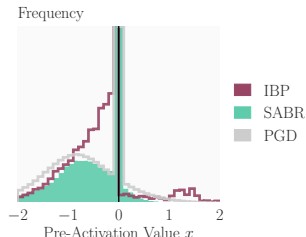

Figure 3: Input distribution for last ReLU layer depending on training method.

For linear layers with weight matrix $\boldsymbol{W}$, we obtain an output radius $\delta^i = |\boldsymbol{W}|\delta^{i-1}$ and thus a constant growth rate $\kappa^i$, corresponding to the row-wise $\ell_1$ norm of the weight matrix $|\boldsymbol{W}_{j,\cdot}|_1$. Empirically, we find most linear and convolutional layers to exhibit growth rates between 10 and 100.

For ReLU layers $\boldsymbol{x}^i = \sigma(\boldsymbol{x}^{i-1})$, the growth rate depends on the location and size of the inputs. Shi et al. (2021) assume the input BOX centers $\bar{\boldsymbol{x}}^{i-1}$ to be symmetrically distributed around 0, i.e., $P_\mathcal{D}(\bar{x}^{i-1}) = P_\mathcal{D}(-\bar{x}^{i-1})$, and obtain a constant growth rate of $\kappa^i = 0.5$. While this assumption holds at initialization, trained networks tend to have more inactive than active ReLUs (see Table 5), indicating asymmetric distributions with more negative inputs (see also Fig. 3). We investigate this more realistic setting. When input radii are $\delta^{i-1} \approx 0$, active neurons will stay stably active, yielding $\delta^i = \delta^{i-1}$ and inactive neurons will stay stably inactive, yielding $\delta^i = 0$. Thus, we obtain a growth rate, equivalent to the portion of active neurons. In the other extreme $\delta^{i-1} \to \infty$, all neurons will become unstable with $\bar{x}^{i-1} \ll \delta^{i-1}$, yielding $\delta^i \approx 0.5\,\delta^{i-1}$, and thus a constant growth rate of $\kappa^i = 0.5$. Assuming pointwise asymmetry favouring negative inputs, i.e., $p(\bar{x}^{i-1} = -z) > p(\bar{x}^{i-1} = z)$, $\forall z \in \mathbb{R}^{>0}$, we show that between those two extremes, output radii grow strictly super-linear in the input size :

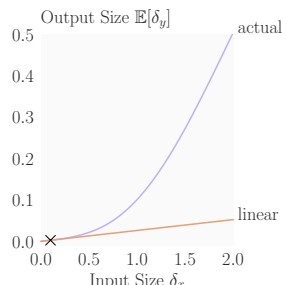

Figure 4: Actual (purple) mean output size growth and a linear approximation (orange) for a ReLU layer with $\bar{x} \sim \mathcal{N}(\mu = -1.0, \sigma = 0.5)$.

**Theorem 4.1** (Hyper-Box Growth). *Let $y := \sigma(x) = \max(0, x)$ be a ReLU function and consider box inputs with radius $\delta_x$ and asymmetrically distributed centers $\bar{x} \sim \mathcal{D}$ such that $P_\mathcal{D}(\bar{x} = -z) > P_\mathcal{D}(\bar{x} = z)$, $\forall z \in \mathbb{R}^{>0}$. Then the mean output radius $\delta_y$ will grow super-linearly in the input radius $\delta_x$. More formally:*

$$\forall \delta_x, \delta_x' \in \mathbb{R}^{\geq 0}: \quad \delta_x' > \delta_x \implies \mathbb{E}_\mathcal{D}[\delta_y'] > \mathbb{E}_\mathcal{D}[\delta_y] + (\delta_x' - \delta_x)\frac{\partial}{\partial \delta_x}\mathbb{E}_\mathcal{D}[\delta_y]. \tag{4}$$

We defer a proof to App. A and illustrate this behavior in Fig. 4. Multiplying all layer-wise growth rates, we obtain the overall growth rate $\kappa = \prod_{i=2}^{L} \kappa^i$, which is exponential in network depth and super-linear in input radius. When not specifically training with the BOX relaxation, we empirically observe that the large growth factors of linear layers dominate the shrinking effect of the ReLU layers, leading to a quick exponential growth in network depth. Further, for both SABR and IBP trained networks, the super-linear growth in input radius empirically manifests as exponential behavior (see Figs. 7 and 8). Using SABR, we thus expect the regularization induced by the robustness term to decrease super-linearly, and empirically even exponentially, with subselection ratio $\lambda$, explaining the significantly higher (standard) accuracies compared to IBP.

## 5 Evaluation

In this section, we evaluate SABR in the challenging $\ell_\infty$-norm setting, deferring a detailed description of the experimental setup to App. B

**Main Results** We compare SABR to state-of-the-art certified training methods in Table 2 and Fig. 5, reporting the best results achieved with a given method on *any* architecture.

In Fig. 5, we show certified over standard accuracy (upper right-hand corner is best) and observe that SABR (◆) dominates all other methods, achieving both the highest certified and standard accuracy across all settings. Methods striving to balance accuracy and regularization by bridging the gap between provable and adversarial training (✖, ●)(Balunovic & Vechev, 2020; Palma et al., 2022) perform only slightly worse than SABR at small perturbation radii, but much worse at large radii, e.g., attaining only 27.5% (✖) and 27.9% (●) certifiable accuracy for CIFAR-10 at $\epsilon = 8/255$ compared to 35.25% (◆). Similarly, methods focusing only on certified accuracy by directly optimizing over-approximations of the worst-case loss (✚, ■) (Gowal et al., 2018b; Zhang et al., 2020) tend to perform well at large perturbation radii, but poorly at small perturbation radii, e.g., on CIFAR-10 at $\epsilon = 2/255$, SABR improves certified accuracy to 62.6% (◆) up from 52.9% (✚) and 54.0% (■).

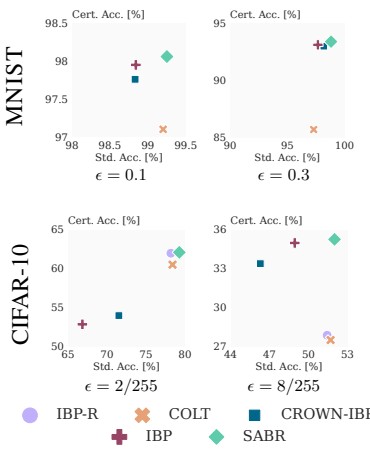

Figure 5: Certified over standard accuracy for different certified training methods. The upper right-hand corner is best.

In contrast to certified training, Zhang et al. (2021) propose an architecture with inherent $\ell_\infty$-robustness properties. While they attain higher certified accuracies on CIFAR-10 $\epsilon = 8/255$, their training is notoriously hard (Zhang et al., 2021, 2022), yielding low standard accuracies of, e.g., only 60.6% compared to 79.52% at $\epsilon = 2/255$. Further, robustness can only be obtained against one perturbation type at a time.

Table 1: Comparison of standard (Std.) and certified (Cert.) accuracy [%] to $\ell_\infty$-distance Net (Zhang et al., 2022).

| Dataset | $\epsilon$ | $\ell_\infty$-distance Net | | SABR (**ours**) | |
|---|---|---|---|---|---|
| | | Std. | Cert. | Std. | Cert. |
| MNIST | 0.1 | 98.93 | 97.95 | **99.25** | **98.06** |
| | 0.3 | 98.56 | 93.20 | **98.82** | **93.38** |
| CIFAR-10 | 2/255 | 60.61 | 54.12 | **79.52** | **62.57** |
| | 8/255 | **54.30** | **40.06** | 52.00 | 35.25 |

**Certification Method and Propagation Region Size** To analyze the interaction between certification method precision and propagation region size, we train a range of models with subselection ratios $\lambda$ varying from 0.0125 to 1.0 and analyze them with verification methods of increasing precision (BOX, DEEPPOLY, MN-BAB) and a 50-step PGD attack (Madry et al., 2018) with 5 random restarts and the targeted logit margin loss (Carlini & Wagner, 2017). We illustrate results in Fig. 6 and observe that standard and adversarial accuracies increase with decreasing $\lambda$, as regularization decreases. For $\lambda = 1$, i.e., IBP training, we

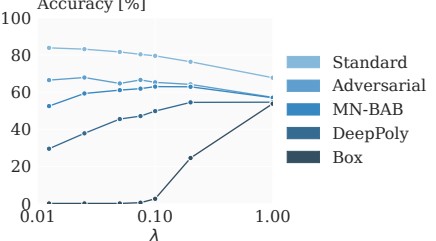

Figure 6: Standard, adversarial, and certified accuracy depending on the certification method for 1000 CIFAR-10 samples at $\epsilon = 2/255$.

observe little difference between the verification methods. However, as we decrease $\lambda$, the BOX verified accuracy decreases quickly, despite BOX relaxations being used during training. In con-

Table 2: Comparison of the standard (Acc.) and certified (Cert. Acc.) accuracy for different certified training methods on the full MNIST, CIFAR-10, and TINYIMAGENET test sets. We use MN-BAB (Ferrari et al., 2022) for certification and report other results from the relevant literature.

| Dataset | $\epsilon_\infty$ | Training Method | Source | Acc. [%] | Cert. Acc. [%] |
|---|---|---|---|---|---|
| MNIST | 0.1 | COLT | Balunovic & Vechev (2020) | 99.2 | 97.1 |
| | | CROWN-IBP | Zhang et al. (2020) | 98.83 | 97.76 |
| | | IBP | Shi et al. (2021) | 98.84 | 97.95 |
| | | SABR | this work | **99.25** | **98.06** |
| | 0.3 | COLT | Balunovic & Vechev (2020) | 97.3 | 85.7 |
| | | CROWN-IBP | Zhang et al. (2020) | 98.18 | 92.98 |
| | | IBP | Shi et al. (2021) | 97.67 | 93.10 |
| | | SABR | this work | **98.82** | **93.38** |
| CIFAR-10 | 2/255 | COLT | Balunovic & Vechev (2020) | 78.4 | 60.5 |
| | | CROWN-IBP | Zhang et al. (2020) | 71.52 | 53.97 |
| | | IBP | Shi et al. (2021) | 66.84 | 52.85 |
| | | IBP-R | Palma et al. (2022) | 78.19 | 61.97 |
| | | SABR | this work | **80.27** | 61.57 |
| | | SABR * | this work | 79.52 | **62.57** |
| | 8/255 | COLT | Balunovic & Vechev (2020) | 51.7 | 27.5 |
| | | CROWN-IBP | Xu et al. (2020) | 46.29 | 33.38 |
| | | IBP | Shi et al. (2021) | 48.94 | 34.97 |
| | | IBP-R | Palma et al. (2022) | 51.43 | 27.87 |
| | | SABR | this work | **52.00** | **35.25** |
| TINYIMAGENET | 1/255 | CROWN-IBP | Shi et al. (2021) | 25.62 | 17.93 |
| | | IBP | Shi et al. (2021) | 25.92 | 17.87 |
| | | SABR | this work | **28.64** | **20.34** |

* With shrinking, see App. B

trast, using the most precise method, MN-BAB, we initially observe increasing certified accuracies, as the reduced regularization yields more accurate networks, before the level of regularization becomes insufficient for certification. While DEEPPOLY loses precision less quickly than BOX, it can not benefit from more accurate networks. This indicates that the increased accuracy, enabled by the reduced regularization, may rely on complex neuron interactions, only captured by MN-BAB.

This qualitatively different behavior depending on the precision of the certification method highlights the importance of recent advances in neural network verification for certified training. Even more importantly, these results clearly show that provably robust networks do not necessarily require the level of regularization introduced by IBP training.

**Loss Analysis** In Fig. 7, we compare the robust loss of a SABR and an IBP trained network across different propagation region sizes (all centered around the original sample) depending on the bound propagation method used. When propagating the full input region ($\lambda = 1$), the SABR trained network yields a much higher robust loss than the IBP trained one. However, when comparing the respective training subselection ratios, $\lambda = 0.05$ for SABR and $\lambda = 1.0$ for IBP, SABR yields significantly smaller training losses, allowing the SABR trained network to reach a much lower standard loss. Finally, we observe the losses to clearly grow super-linearly with increasing propagation region sizes (note the logarithmic scaling of the y-axis) agreeing well with our theoretical results in §4.

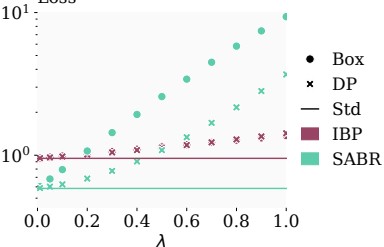

Figure 7: Standard and robust cross-entropy loss, computed with BOX (Box) and DEEPPOLY (DP) bounds for an IBP and SABR trained network over subselection ratio $\lambda$.

## 6 Conclusion

We introduced a novel certified training method called SABR (**S**mall **A**dversarial **B**ounding **R**egions) based on the key insight, that propagating small but carefully selected subsets of the input region combines small approximation errors and thus regularization with well-behaved optimization problems. This allows SABR trained networks to outperform *all* existing certified training methods on *all* commonly used benchmarks in terms of *both* standard and certified accuracy. Even more importantly, SABR lays the foundation for a new class of certified training methods promising to overcome the robustness-accuracy trade-off and enabling the training of networks that are both accurate and certifiably robust.

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

# A Deferred Proofs

In this section, we provide the proof for Lemma A.1. Let us first consider the following Lemma:

**Lemma A.1** (Hyper-Box Growth). *Let $y := \sigma(x) = \max(0, x)$ be a ReLU function and consider box inputs with radius $\delta_x$ and centers $\bar{x} \sim \mathcal{D}$. Then the mean radius $\mathbb{E}\delta_y$ of the output boxes will satisfy:*

$$\frac{\partial}{\partial \delta_{x,i}} \mathbb{E}_{\mathcal{D}}[\delta_{y,i}] = \frac{1}{2} P_{\mathcal{D}}[-\delta_{x,i} < \bar{x}_i < \delta_{x,i}] + P_{\mathcal{D}}[\bar{x}_i > \delta_{x,i}] > 0, \tag{5}$$

*and*

$$\frac{\partial}{\partial^2 \delta_{x,i}} \mathbb{E}_{\mathcal{D}}[\delta_{y,i}] = \frac{1}{2}(P_{\mathcal{D}}[\bar{x}_i = -\delta_{x,i}] - P_{\mathcal{D}}[\bar{x}_i = \delta_{x,i}]). \tag{6}$$

*Proof.* Recall that given an input box with center $\bar{x}$ and radius $\delta_x$, the output relaxation of a ReLU layer is defined by:

$$\bar{y}_i = \begin{cases} 0, & \text{if } \bar{x}_i + \delta_{x,i} \leq 0 \\ \frac{\bar{x}_i + \delta_{x,i}}{2}, & \text{elif } \bar{x}_i - \delta_{x,i} \leq 0 \\ \bar{x}_i, & \text{else} \end{cases}, \qquad \delta_{y,i} = \begin{cases} 0, & \text{if } \bar{x}_i + \delta_{x,i} \leq 0 \\ \frac{\bar{x}_i + \delta_{x,i}}{2}, & \text{elif } \bar{x}_i - \delta_{x,i} \leq 0 \\ \delta_{x,i}, & \text{else} \end{cases} \tag{7}$$

We thus obtain the expectation

$$\mathbb{E}_{\mathcal{D}}[\delta_{y,i}] = \int_{-\delta_{x,i}}^{\delta_{x,i}} \frac{\bar{x}_i + \delta_{x,i}}{2} p[\bar{x}_i] d\bar{x}_i + \int_{\delta_{x,i}}^{\infty} \delta_{x,i} p_D(\bar{x}_i) d\bar{x}_i$$

$$= \frac{\delta_{x,i}}{2} P_{\mathcal{D}}[-\delta_{x,i} < \bar{x}_i < \delta_{x,i}] + \delta_{x,i} P_{\mathcal{D}}[\bar{x}_i > \delta_{x,i}] + \int_{-\delta_{x,i}}^{\delta_{x,i}} \frac{\bar{x}_i}{2} p[\bar{x}_i] d\bar{x}_i, \tag{8}$$

its derivative

$$\frac{\partial}{\partial \delta_{x,i}} \mathbb{E}_{\mathcal{D}}[\delta_{y,i}] = \frac{1}{2} P_{\mathcal{D}}[-\delta_{x,i} < \bar{x}_i < \delta_{x,i}] + \frac{\delta_{x,i}}{2}(P_{\mathcal{D}}[\bar{x}_i = -\delta_{x,i}] + P_{\mathcal{D}}[\bar{x}_i = \delta_{x,i}])$$

$$+ P_{\mathcal{D}}[\bar{x}_i > \delta_{x,i}] - \delta_{x,i} P_{\mathcal{D}}[\bar{x}_i = \delta_{x,i}]$$

$$+ \frac{\delta_{x,i}}{2}(P_{\mathcal{D}}[\bar{x}_i = -\delta_{x,i}] - P_{\mathcal{D}}[\bar{x}_i = \delta_{x,i}])$$

$$= \frac{1}{2} P_{\mathcal{D}}[-\delta_{x,i} < \bar{x}_i < \delta_{x,i}] + P_{\mathcal{D}}[\bar{x}_i > \delta_{x,i}] > 0, \tag{9}$$

and its curvature

$$\frac{\partial}{\partial^2 \delta_{x,i}} \mathbb{E}_{\mathcal{D}}[\delta_{y,i}] = \frac{1}{2}(P_{\mathcal{D}}[\bar{x}_i = -\delta_{x,i}] + P_{\mathcal{D}}[\bar{x}_i = \delta_{x,i}]) - P_{\mathcal{D}}[\bar{x}_i = \delta_{x,i}]$$

$$= \frac{1}{2}(P_{\mathcal{D}}[\bar{x}_i = -\delta_{x,i}] - P_{\mathcal{D}}[\bar{x}_i = \delta_{x,i}]). \tag{10}$$

$\square$

Now, we can easily proof Theorem 4.1, restated below for convenience.

**Theorem 4.1** (Hyper-Box Growth). *Let $y := \sigma(x) = \max(0, x)$ be a ReLU function and consider box inputs with radius $\delta_x$ and asymmetrically distributed centers $\bar{x} \sim \mathcal{D}$ such that $P_{\mathcal{D}}(\bar{x} = -z) > P_{\mathcal{D}}(\bar{x} = z)$, $\forall z \in \mathbb{R}^{>0}$. Then the mean output radius $\delta_y$ will grow super-linearly in the input radius $\delta_x$. More formally:*

$$\forall \delta_x, \delta'_x \in \mathbb{R}^{\geq 0}: \quad \delta'_x > \delta_x \implies \mathbb{E}_{\mathcal{D}}[\delta'_y] > \mathbb{E}_{\mathcal{D}}[\delta_y] + (\delta'_x - \delta_x)\frac{\partial}{\partial \delta_x}\mathbb{E}_{\mathcal{D}}[\delta_y]. \tag{4}$$

*Proof.* We apply Lemma A.1 by substituting an asymmetric center distribution $\mathcal{D}$, satisfying $P_{\mathcal{D}}(\bar{x} = -z) > P_{\mathcal{D}}(\bar{x} = z)$, $\forall z \in \mathbb{R}^{>0}$ into Eq. (6) to obtain:

$$\frac{\partial}{\partial^2 \delta_{x,i}} \mathbb{E}_{\mathcal{D}}[\delta_{y,i}] = \frac{1}{2}(P_{\mathcal{D}}[\bar{x}_i = -\delta_{x,i}] - P_{\mathcal{D}}[\bar{x}_i = \delta_{x,i}]) > 0.$$

The theorem follows trivially from the strictly positive curvature. $\square$

**Example for Piecewise Uniform Distribution** Let us assume the centers $\bar{x} \sim D$ are distributed according to:

$$P_{\mathcal{D}}[\bar{x} = z] = \begin{cases} a, & \text{if } -l \leq z < 0 \\ b, & \text{elif } 0 < u \leq l \\ 0, & \text{else} \end{cases}, \quad l = \frac{1}{a+b}, \tag{11}$$

where $a$ and $b$. Then we have by Lemma A.1

$$\mathbb{E}_{\mathcal{D}}[\delta_y] = \frac{\delta_x}{2} P_{\mathcal{D}}[-\delta_x < \bar{x} < \delta_x] + \delta_x P_{\mathcal{D}}[\bar{x} > \delta_x] + \int_{-\delta_x}^{\delta_x} \frac{\bar{x}}{2} p[\bar{x}] d\bar{x} \tag{12}$$

$$= \frac{\delta_x^2}{2}(a + b) + b\delta_x(l - \delta_x) + \frac{\delta_x^2}{4}(b - a) \tag{13}$$

$$= \delta_x^2 \frac{a-b}{4} + \delta_x \frac{b}{a+b}. \tag{14}$$

We observe quadratic growth for $a > b$ and recover the symmetric special case of $\mathbb{E}_{\mathcal{D}}[\delta_y] = 0.5\delta_x$ for $a = b$.

# B  Additional Experimental Details

In this section, we provide detailed informations on the exact experimental setup.

**Experimental Setup** We implement SABR in PyTorch (Paszke et al., 2019) and use MN-BAB (Ferrari et al., 2022) for certification. We conduct experiments on MNIST (LeCun et al., 2010), CIFAR-10 (Krizhevsky et al., 2009), and TINYIMAGENET (Le & Yang, 2015) for the challenging $\ell_\infty$ perturbations, using the same 7-layer convolutional architecture CNN7 as prior work (Shi et al., 2021) unless indicated otherwise (see App. B for more details). We choose similar training hyperparameters as prior work (Shi et al., 2021) and provide more detailed information in App. B.

**Datasets** We conduct experiments on the MNIST (LeCun et al., 2010), CIFAR-10 (Krizhevsky et al., 2009), and TINYIMAGENET (Le & Yang, 2015) datasets. For TINYIMAGENET and CIFAR-10 we follow Shi et al. (2021) and use random horizontal flips and random cropping as data augmentation during training and normalize inputs after applying perturbations. Following prior work (Xu et al., 2020; Shi et al., 2021), we evaluate CIFAR-10 and MNIST on their test sets and TINY-IMAGENET on its validation set, as test set labels are unavailable. Following Xu et al. (2020) and in contrast to Shi et al. (2021), we train and evaluate TINYIMAGENET with images cropped to $56 \times 56$.

**Training Hyperparameters** We mostly follow the hyperparameter choices from Shi et al. (2021) including their weight initialization and warm-up regularization[2], and use ADAM (Kingma & Ba, 2015) with an initial learning rate of $5 \times 10^{-4}$, decayed twice with a factor of 0.2. For CIFAR-10 we train 160 an 180 epochs for $\epsilon = 2/255$ and $\epsilon = 8/255$, respectively, decaying the learning rate after 120 and 140 and 140 and 160 epochs. For TINYIMAGENET $\epsilon = 1/255$ we use the same settings as for CIFAR-10 at $\epsilon = 8/255$. For MNIST we train 70 epochs, decaying the learning rate after 50 and 60 epochs. We

Table 3: Hyperparameters for the experiments shown in Table 2.

| Dataset | $\epsilon$ | $\ell_1$ | $\lambda$ |
|---|---|---|---|
| MNIST | 0.1 | $10^{-5}$ | 0.4 |
| | 0.3 | $10^{-6}$ | 0.6 |
| CIFAR-10 | 2/255 | $10^{-6}$ | 0.1 |
| | 8/255 | 0 | 0.7 |
| TINYIMAGENET | 1/255 | $10^{-6}$ | 0.4 |

choose a batch size of 128 for CIFAR-10 and TINYIMAGENET, and 256 for MNIST. We use $\ell_1$ regularization with factors according to Table 3. For all datasets, we perform one epoch of standard training ($\epsilon = 0$) before annealing $\epsilon$ from 0 to its final value over 80 epochs for CIFAR-10 and TINYIMAGENET and for 20 epochs for MNIST. We use an $n = 8$ step PGD attack with an initial step size of $\alpha = 0.5$, decayed with a factor of 0.1 after the 4[th] and 7[th] step to select the center of the propagation region. We use a constant subselection ratio $\lambda$ with values shown in Table 3. For CIFAR-10 $\epsilon = 2/255$ we use shrinking with $c_s = 0.8$ (see below).

---

[2]For the ReLU warm-up regularization, the bounds of the small boxes are considered.

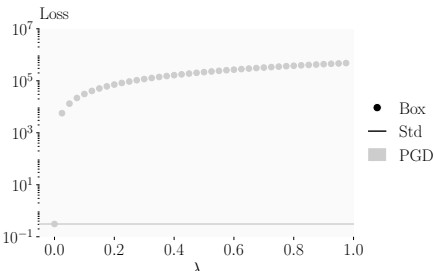 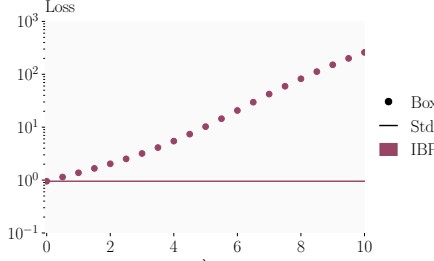

Figure 8: Standard (Std.) and robust cross-entropy loss, computed with BOX (Box) bounds for an adversarially (left) and IBP (right) trained network over subselection ratio $\lambda$. Note the logarithmic y-scale and different axes.

**ReLU-Transformer with Shrinking**    Additionally to standard SABR, outlined in §3, we propose to amplify the BOX growth rate reduction (see §4) affected by smaller propagation regions, by adapting the ReLU transformer as follows:

$$\bar{y}_i = \begin{cases} 0, & \text{if } \bar{x}_i + \delta_{x,i} \leq 0 \\ c_s \frac{\bar{x}_i + \delta_{x,i}}{2}, & \text{elif } \bar{x}_i - \delta_{x,i} \leq 0 \text{ ,} \\ \bar{x}_i, & \text{else} \end{cases} \qquad \delta_{y,i} = \begin{cases} 0, & \text{if } \bar{x}_i + \delta_{x,i} \leq 0 \\ c_s \frac{\bar{x}_i + \delta_{x,i}}{2}, & \text{elif } \bar{x}_i - \delta_{x,i} \leq 0 \text{ .} \\ \delta_{x,i}, & \text{else} \end{cases} \qquad (15)$$

We call $c_s$ the shrinking coefficient, as the output radius of unstable ReLUs is shrunken by multiplying it with this factor. We only use these transformers for the marked $(*)$ CIFAR-10 $\epsilon = 2/255$ network discussed in Table 2.

**Architectures**    Similar to prior work (Shi et al., 2021), we use a 7-layer convolutional architecture, CNN7. The first 5 layers are convolutional layers with filter sizes [64, 64, 128, 128, 128], kernel size 3, strides [1, 1, 2, 1, 1], and padding 1. They are followed by a fully connected layer with 512 hidden units and the final classification. All but the last layers are followed by batch normalization (Ioffe & Szegedy, 2015) and ReLU activations. For the BN layers, we train using the statistics of the unperturbed data similar to Shi et al. (2021). During the PGD attack we use the BN layers in evaluation mode. We further consider narrower version, CNN7-narrow which is identical to CNN7 expect for using the filter sizes [32, 32, 64, 64, 64] and a fully connected layer with 216 hidden units.

**Hardware and Timings**    We train and certify all networks using single NVIDIA RTX 2080Ti, 3090, Titan RTX, or A6000. Training takes roughly 3 and 7 hours for MNIST and CIFAR-10, respectively, with TINYIMAGENET taking two and a half days on a single NVIDIA RTX 2080Ti. For more Details see Table 4. Verification with MN-BAB takes around 34h for MNIST, 28h for CIFAR-10 and 2h for TINYIMAGENET on a NVIDIA Titan RTX.

Table 4: SABR training times on a single NVIDIA RTX 2080Ti.

| Dataset | $\epsilon$ | Time |
|---|---|---|
| MNIST | 0.1 | 3h 23 min |
| | 0.3 | 3h 20 min |
| CIFAR-10 | 2/255 | 7h 6 min |
| | 8/255 | 7h 20 min |
| TINYIMAGENET | 1/255 | 57h 24 min |

## C  Additional Experimental Results

**Loss Analysis**    In Fig. 8, we show the error growth of an adversarially trained (left) and IBP trained model over increasing subselection ratios $\lambda$. We observe that errors grow only slightly super-linear rather than exponential for the adversarially trained network. We trace this back to the large portion of crossing ReLUs (Table 5), especially in later layers, leading to the layer-wise growth being only linear. For the IBP trained model, in contrast, we observe exponential growth across a wide range of propagation region sizes, as the heavy regularization leads to a

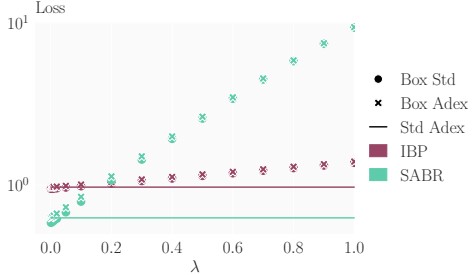

Figure 9: Comparison of the robust cross-entropy losses computed with BOX (Box) centered around unperturbed and adversarial examples for an IBP and SABR trained network over subselection ratio $\lambda$.

small portion of active and unstable ReLUs. In Fig. 9, we compare BOX errors around the unperturbed sample and the center computed with an adversarial attack, as described in §3. We observe that while the loss is larger around the adversarial centers, especially for small propagation regions, this effect is small compared to the difference between training or certification methods.

**ReLU Activation States**  The portion of ReLU activations which are (stably) active, inactive, or unstable has been identified as an important characteristic of certifiably trained networks (Shi et al., 2021). We evaluate these metrics for IBP, SABR, and adversarially (PGD) trained networks on CIFAR-10 at $\epsilon = 2/255$, using the BOX relaxation to compute intermediate bounds, and report the average over all layers and test set samples in Table 5. We observe that, when evaluated on concrete

Table 5: Average percentage of active, inactive, and unstable ReLUs for concrete points and boxes depending on training method.

| Method | Point | | Whole Region | | |
|---|---|---|---|---|---|
| | Act | Inact | Unst | Act | Inact |
| IBP | 26.2 | 73.8 | 1.18 | 25.6 | 73.2 |
| SABR | 35.9 | 64.1 | 3.67 | 34.3 | 62.0 |
| PGD | 36.5 | 63.5 | 65.5 | 15.2 | 19.3 |

points, the SABR trained network has around 37% more active ReLUs than the IBP trained one and almost as many as the PGD trained one, indicating a significantly smaller level of regularization. While the SABR trained network has around 3-times as many unstable ReLUs as the IBP trained network, when evaluated on the whole input region, it has 20-times fewer than the PGD trained one, highlighting the improved certifiability.

**Gradient Alignment**  To analyze whether SABR training is actually more aligned with standard accuracy and empirical robustness, as indicated by our theory in §4, we conduct the following experiment for CIFAR-10 and $\epsilon = 2/255$: We train one network using SABR with $\lambda = 0.05$ and one with IBP, corresponding to $\lambda = 1.0$. For both, we now compute the gradients $\nabla_\theta$ of their respective robust training losses $\mathcal{L}_{\text{rob}}$ and the cross-entropy loss $\mathcal{L}_{\text{CE}}$ applied to unperturbed (Std.) and adversarial (Adv.) samples. We then report the mean cosine similarity between these gradients across the whole test set in Table 6. We clearly observe that the SABR loss is much better aligned with both the cross-entropy loss of unperturbed and adversarial samples, corresponding to standard accuracy and empirical robustness, respectively.

Table 6: Cosine similarity between $\nabla_\theta \mathcal{L}_{\text{rob}}$ for IBP and SABR and $\nabla_\theta \mathcal{L}_{\text{CE}}$ for adversarial (Adv.) and unperturbed (Std.) examples.

| Loss | IBP | SABR |
|---|---|---|
| Std. | 0.5586 | **0.8071** |
| Adv. | 0.8047 | **0.9062** |

