# OpenReview forum: "Certified Training: Small Boxes are All You Need"
_NeurIPS.cc/2022/Workshop/TSRML — TSRML2022_

### Official Review · Reviewer_dwMh · 2022-10-11
**Interesting method achieving great results on Certified adversarial robustness.**

**Overall Rating:** 9

**Summary:**

This paper introduces a new training method to achieve robustness to small L_inf perturbations.

The idea consists in propagating *smaller regions* than the one we want to prove robustness for, but to propagate them around the points that are more likely to be violations of the specification. An adversarially bad point is found using a PGD attack, and the loss is then computed around a small region (which is a fraction of the size of the original specification region) that will contain this point.

As a result, because the bounds propagated are much smaller, the amount of over-approximation happening is less than using the usual certified training. This will cause the network to be less over-regularized, and leads to the ability to achieve better standard accuracy.


**Strengths:**

This is a very nice paper, with some definitely novel ideas that are challenging the usual way that robustness training is done (assuming that you always need to optimize an upper bound of the expected worst case loss). The presentation is very clear, with Figure 1 giving a good overview of the motivation, while Figure 2 explaining clearly the actual procedure used for training.
Empirical evaluation is performed over the usual benchmarks on L_inf robustness, where significant improvements, notably in terms of standard accuracy, are achieved.

**Weaknesses:**

# Questions / Suggestions

## Suggestion to the authors:
The suggested approach rely on training with *smaller* epsilon than is going to be used for certification. This is an interesting opposition to a standard trick that is used in other methods (off the top of my head, at least in IBP-R and in COLT) to train with a *larger* epsilon, with the intuition being that over-regularizing makes it more likely for the robustness of the network to generalize to unseen points. Given the diametrically opposed strategies, a discussion of this might be of interest.

##  Question about training:
One disadvantage about the proposed method is that, unlike the usual certified training, the loss that is used for the training objective is not an upper bound on the loss that we care about anymore. As a result, it become possible for the optimizer to improve the surrogate loss (small bounds propagated around adversarial example) rather than the one we actually care about. You could think for example about  the PGD attack getting stuck due to some gradient masking, and therefore the bound being computed only around certain area which might not be the ones with highest adversarial loss. It seems to not be a problem given the good empirical results achieved but I was wondering if the authors found some difficulty in tuning hyperparameters of the PGD attack or if the method was relatively robust wrt hyperparameters choice?

## Question about verification:
Given that the point of the proposed method is to lower the amount of regularization so as to improve standard accuracy, did the authors observe an impact on verification time when compared to other methods? I would expect more regularized networks to be faster to verify, but am unsure if this intuition is supported by the data?

# Criticism

## Lack of scale for Figure 5.
There is no scale in the x and y axis of the graphs in Figure 5. I understand that the numbers can be found in Table 2 but as it is, the figure is just confusing.

## Relu Transformer with Shrinking (Appending l.392 to 397)
I would recommend to the authors to avoid changing the method that they are using just for a single benchmark. If shrinking the output of the ReLU is an important part of the method, put it in the main text, apply it to all networks and report results on that. As it is, what this looks like is "We will do random tweaks to our training until we get one network that beats our baselines and then report that". It is acceptable for a method to not be the best on all benchmarks, and it is much more valuable to have clear numbers for evaluation of each method.

**Overall Recommendation:**

Definitely recommend accepting, the paper proposes a well-explained, simple idea, that seems to be giving great results in practice. I suggested some possible improvements / points worth discussing to the authors but the paper is quite good as is.

**Review Confidence:**

5: The reviewer is absolutely certain that the evaluation is correct and very familiar with the relevant literature

---

### Official Review · Reviewer_r1qu · 2022-10-20
**Small Boxes Might be All You Need**

**Overall Rating:** 8

**Summary:**

The paper suggests a novel approach to adversarial training. The paper combines two approaches for solving the inner optimization problem for adversarial training: adversarial attack-based point estimation approaches that use an adversarial attack to find an adversarial example and certification region-based approaches that propagate the input region to find a local linear bound on the logit differences. The new method SABR propagates a smaller box that contains an adversarial example.

The authors suggest that the new approach using smaller boxes gives better stability than point estimate-based methods, as they ensure robustness over a region. They are also better than IBP-like approaches as propagating a smaller region gives tighter relaxations without having too many unstable ReLUs in the certification region. The authors also provide theoretical and empirical justification for the claims mentioned above.

**Strengths:**

- The suggested idea is novel and would be very relevant to the community.
- The authors also provide some theoretical justification for the approach's success.
- The presentation of the material is clear and easy to understand.

**Weaknesses:**

The scope of the threat models considered in the paper is limited. The approach could be extended to other lp norms.
Moreover, in the $\ell_\infty$ threat model, the authors could also consider more cuboidal box constraints for the small boxes.

**Overall Recommendation:**

I think the work provides an exciting idea for improving adversarial training. The empirical improvements in the paper are pretty decent, and I think the idea could be optimized and developed to train even more robust models. I recommend acceptance.

**Review Confidence:**

4: The reviewer is confident but not absolutely certain that the evaluation is correct

---

### Official Review · Reviewer_2pj5 · 2022-10-21
**can be improved**

**Overall Rating:** 7

**Summary:**

The paper proposed a new certified training method by using smaller subset of the adversarial input region to approximate the worst-case loss. Experiments on MIST, CIFAR, and TinyImagenet validate the usefulness of the proposed SABR.

**Strengths:**

The paper proposes an interesting way of improving certified training performance by exploring the fact that a small subregion in the adversarial input region is sufficient to approximate the worst-case loss. The characteristics of the robustness loss and growth of small boxes are also beneficial for understanding the need for SABR.

**Weaknesses:**

While the paper is generally easy to follow, there are several parts that are confusing. For example, Figure 2 is not clear and does not explain the procedures well. In Main Results, it is said that the best results on any architecture with a given method are reported, without specifying the range of architectures they considered. Also, how should one select the proper $\lambda$.

**Overall Recommendation:**

Although the paper still has many problems (see above), it has good contribution to the certified training algorithms.

**Review Confidence:**

2: The reviewer is willing to defend the evaluation, but it is quite likely that the reviewer did not understand central parts of the paper

---

### Decision · Program_Chairs · 2022-10-23

**Decision:**

Accept

**Comment:**

Following the unanimous recommendations from reviewers, the submission is accepted.